# A Spoof Surface Plasmon Polaritons (SSPPs) Based Dual-Band-Rejection Filter with Wide Rejection Bandwidth

**DOI:** 10.3390/s20247311

**Published:** 2020-12-19

**Authors:** Ehsan Farokhipour, Mohammad Mehrabi, Nader Komjani, Can Ding

**Affiliations:** 1Department of Electrical Engineering, Iran University of Science and Technology, Tehran 1684613114, Iran; ehsan_farokhi@alumni.iust.ac.ir (E.F.); mohmg@kth.se (M.M.); 2Global Big Data Technologies Centre, University of Technology Sydney, Sydney, NSW 2007, Australia; Can.Ding@uts.edu.au

**Keywords:** tunability, band stop filter, spoof surface plasmon polaritons, circular ring resonators

## Abstract

This paper presents a novel single-layer dual band-rejection-filter based on Spoof Surface Plasmon Polaritons (SSPPs). The filter consists of an SSPP-based transmission line, as well as six coupled circular ring resonators (CCRRs) etched among ground planes of the center corrugated strip. These resonators are excited by electric-field of the SSPP structure. The added ground on both sides of the strip yields tighter electromagnetic fields and improves the filter performance at lower frequencies. By removing flaring ground in comparison to prevalent SSPP-based constructions, the total size of the filter is significantly decreased, and mode conversion efficiency at the transition from co-planar waveguide (CPW) to the SSPP line is increased. The proposed filter possesses tunable rejection bandwidth, wide stop bands, and a variety of different parameters to adjust the forbidden bands and the filter’s cut-off frequency. To demonstrate the filter tunability, the effect of different elements like number (*n*), width ($W_{R}$), radius ($R_{R}$) of CCRRs, and their distance to the SSPP line ($y_{R}$) are surveyed. Two forbidden bands, located in the X and K bands, are 8.6–11.2 GHz and 20–21.8 GHz. As the proof-of-concept, the proposed filter was fabricated, and a good agreement between the simulation and experiment results was achieved.

## 1. Introduction

Surface plasmon polaritons (SPPs) have the ability to miniaturize the photonic components and overcome the classical diffraction limit with scales smaller than the light-line. The mentioned properties have made SPPs promising in huge potential applications of super-resolution imaging, bio-sensing, and near field microscopy. SPPs do not exist naturally in the microwave and THz frequency regime, so plasmonic metamaterials (PMMs) containing periodic sub-wavelength cells were proposed in the name of Spoof SPPs (SSPPs). PMMs are advancing at a rapid pace, and are considered as an interesting candidate in the practical applications [1,2]. The SSPP modes are a kind of special surface electromagnetic waves that are highly localized to the dielectric-metal interface and exponentially decay away from the interface. SSPPs have several advantages, such as supporting bend structures and flexible substrate, being naturally broadband, low fabrication cost, low cross-coupling, and lower ohmic & dielectric loss, in comparison to conventional technologies. Moreover, they have controllable properties by adjusting geometrical parameters [3,4,5]. In recent years, SSPPs have become a research hotspot and have been extensively studied in different functional equipment, such as transmission lines (TL) [6,7], power dividers [8], couplers [9], sensors [10], antennas [11], and filters [12,13,14,15,16,17,18,19,20].

A compact band-stop filter that obstructs desired frequency spectrums is a pivotal device for modern communication systems to reduce noise and suppress interface. Besides, the ability to adjust the rejection-band is demandable in advanced smart systems. Specifically, the intrinsic property of SSPPs, which have controllable characteristics by simply adjusting the geometrical parameters, facilitates miniaturized filtering designs, especially tunable multi-band-rejection filters (MBRF). Compared to the traditional types of filters based on microstrip [21,22,23,24,25] or substrate integrated waveguide [26,27,28], SSPP-based filters have a potential to attain lower loss, smaller size, and less cross coupling [29].

There exist various MBRF designs based on SSPPs in the literature [30,31,32,33,34,35,36,37]. In Reference [30], by etching split-ring resonators (SRRs) on an SSPP waveguide, a MBRF with two narrow band rejections at 8.21 GHz and 10.04 GHz is obtained. In Reference [31], more than 20 dB rejections at 5.36 GHz and 9.32 GHz are achieved with 10 dB fractional bandwidth 1.07 and 0.74%, respectively. In Reference [32], around 8 dB rejections with center frequencies 4.04 and 4.5 GHz are generated by introducing multiple defect units into an ultra-thin periodic corrugated metallic strip. Ref. [33] introduces the MBRFs based on the interaction between an SSPP-TL and several ring resonators. Two narrow rejections at 7.65 and 9.47 GHz is obtained by using two rows of electric-field-coupled LC (ELC) particles on each side of a meliorated SSPP waveguide with two different ELC scaling coefficients. Tian et al, in Reference [34], present a double-layer SSPP-based MBRF at 5, 12.5, and 22 GHz by using stop bands among the dispersion modes of the SSPP unit-cell. In this study, there is no discussion of how to control the forbidden bands. In Reference [35], a dual-BRF is attained through adding interdigital capacitance loaded loop resonators (IDCLLRs) into the grooves of the corrugated SSPP-based transmission line. The first rejection band is from 8.9 to 9.4 GHz with the relative bandwidth of 5.5%, while the other stop-band is from 11 to 12.1 GHz with the relative bandwidth of 9.5%. In addition, the designed double-layer MBRF in Reference [36] is realized by cascading SSPP units of the same structures but with diverse rejections bands. The first stop-band possesses a bandwidth of 0.96 GHz at a center frequency of 18.6 GHz with the relative bandwidth of 5.2%, while the other one has a relative bandwidth of 6% at the center frequency of 22.6 GHz. In Reference [37], Aziz et al. introduce a new methodology to realize a high-performance MBRF at 6.4 and 6.8 GHz through etching metamaterial particles on the SSPP waveguide. The forbidden bands can be adjusted by simply changing the metamaterial decorations. Unfortunately, the proposed filter possesses narrow forbidden bands and large dimensions. The total size of the filter is 502 mm * 65.65 mm, which is 11.15$\lambda_{g}$ * 1.44$\lambda_{g}$ ($\lambda_{g}$ is the guided wavelength at frequency around the average center frequency of the forbidden bands). The aforementioned MBRFs in the above literature all have narrow rejection bandwidths, usually less than 10%. However, wider stop bands are required in some scenarios, e.g., when the MBRF is connected with an ultra-wide-band receiving antenna to mitigate the interferences from various satellites working at different spectrums.

In this paper, an SSPP-based dual-band-rejection filter (DBRF) is proposed. The filter has two wide forbidden bands at X and K bands with fractional bandwidths of 26.2% and 8.6%, respectively. The forbidden bands can be easily adjusted by varying its geometrical parameters. In addition, the filter has only one metal layer, which has a good flexibility for conformal designs. The process of realizing the filter is as follows: (1) designing a low-pass filter to determine the DBRF’s high cut-off frequency; (2) projecting the DBRF by electric-field interaction between SSPP-TL and CCRRs; and (3) a parametric study to investigate the filter tunability. The filter’s high cut-off frequency and the forbidden bands can be independently controlled by tuning the SSPP unit-cell and the geometrical parameters in the filter part, respectively. Using such a structure with ground planes on double sides of the corrugated strip engenders tighter EM fields and improves the performance of the filter at lower frequencies. Eventually, experiments are conducted to confirm the excellent filtering performance of the structure.

## 2. Filter Design

### 2.1. Principle Mechanism

The schematic configuration of the proposed DBRF is presented in Figure 1. The DBRF has one metallic layer (yellow) and one dielectric layer (blue). The design is based on Rogers 4003C PCB board with the thickness of 0.208 mm, the permittivity of 3.55, and the loss tangent of tanδ = 0.0027. The proposed DBRF consists of five sections, i.e., one main filtering section, including a corrugated SSPP-based line and six coupled circular ring resonators (CCRRs), two transition sections made of co-planar waveguide (CPW) line with gradient corrugated grooves, and two feed ports which are conventional 50-Ohm CPW lines. The port sections, the transition sections, and the filtering section are labelled as Regions I, II, and III in Figure 1. The 50-Ohm CPW-line in Region I has a line width, gap width, and length of *H* = 2.9 mm, *g* = 0.3 mm, and $L_{1}$ = 2.65 mm. The period of the CPW-based H-shaped cells in Regions II and III is *P* = 1.95 mm. In Region II, the gradient grooves have uniform width of *b* = 0.65 mm but their depths increase from $d_{1}$ = 0.195 mm to $d_{5}$ = 0.975 mm with a step of 0.195 mm. The length of each transition section is $L_{2}$ = 5 * *P* = 9.75 mm. The transition sections are employed to provide a good matching between the filtering section and the 50-Ohm conventional CPW lines. Note that slight variations of the gradient groove parameters could lead to a better matching, but with a longer length. A trade-off between the size and the matching was made when optimizing the transition sections. In Region III, uniform grooves with depth of *d* = $d_{5}$ = 0.975 mm are used in the CPW line to excite the SSPP. Moreover, six CCRRs are etched on ground plane of the SSPP line. The width and radius of the CCRRs are $W_{R}$ = 0.2 mm, $R_{R}$ = 2.4 mm, respectively; the distance between adjacent CCRRs is *D* = 5.55 mm; the distance between the CCRRs and the SSPP line is $y_{R}$ = 3.75 mm. All the simulations were conducted with a commercial software CST Studio Suit 2020 [38]. During simulations, frequency domain solver with tetrahedral mesh was used.

### 2.2. SSPP Unit-Cell Analysis

To excite the SSPP, as shown in Figure 1, H-shaped sub-wavelength unit cells are periodically arrayed on the CPW line to bind the electromagnetic fields to the interface. The dispersion characteristic of the unit cells with different groove depths d is depicted in Figure 2a. As can be observed from the figure, the dispersion curve is under the light line (when *d* = 0), indicating a slow wave propagation. With a lager d, the cut-off frequency gets lower. To demonstrate that the SSPP is successfully excited by the H-shaped unit cells with *d* = 0.975 mm, the E-field distribution around the interface of the metal and dielectric and the H-field distribution on the x-y plane are presented in Figure 2c. A zoom-in view of the H-shaped unit cell is also included in Figure 2c to facilitate the understanding of the field distribution. It can be observed from Figure 2c that the SSPP is excited at the interface of the dielectric and metal of the H-shaped unit cell. The bounded SSPP modes demonstrate an exponential attenuation away from the interface in the vertical direction.

The relationship between the wave number in free space and the propagation constants in different directions can be expressed as $k_{0}^{2}=k_{x}^{2}+k_{y}^{2}+k_{z}^{2}$. According to the Maxwell Equations, the EM fields should satisfy the following equations:(1)$$E_{x}=E_{0}e^{\left(-jk_{y}y\right)}e^{\left(-jk_{z}z\right)}e^{\left(-jk_{x}x\right)},$$
(2)$$H_{z}=-\frac{\omega \varepsilon k_{y}}{\left(k_{y}^{2}+k_{z}^{2}\right)}E_{0}e^{\left(-jk_{y}y\right)}e^{\left(-jk_{z}z\right)}e^{\left(-jk_{x}x\right)}.$$

The surface impedance along the +x direction is computed as:(3)$$\eta_{surf}\left(x\right)=\frac{E_{x}}{H_{z}}=-\frac{\left(k_{y}^{2}+k_{z}^{2}\right)}{\omega \varepsilon k_{y}}\frac{\eta_{0}}{k_{0}},$$
where $\eta_{0}$ is the wave impedance in the free space (377-Ohm). Since the surface waves decay exponentially in vertical direction to the interface, we have $k_{y}\gg k_{z}$. According to the dispersion characteristics of the unit-cell shown in Figure 2a, $k_{x}$ can be changed by adjusting the groove depth. Therefore, the surface impedance along the propagation direction of each cell can be calculated as:(4)$$\eta_{surf}\left(x\right)=\eta_{0}\sqrt{1-{\left(\frac{k_{x}}{k_{0}}\right)}^{2}}.$$

The surface impedance of each unit cell of the transmission line at different frequencies with different groove depths are plotted in Figure 2b. For example, at 14.5 GHz, the surface impedance of the plasmonic transmission line can cover the range of 380 to 525 when *d* varies from 0.195 to 0.975 mm. It should be noted that the surface impedance in the filtering section will be modulated by adding circular resonators; thus, band stops will be created in the transmission spectrum of the SSPP-based transmission line. In addition, changing the geometric parameters of the resonators leads to a variation of the local surface impedance in the filtering part, as well. So, having tunable stop bands will be expected.

### 2.3. Theoretical Study of the CCRRs

According to the dispersion diagram of the SSPP-cell, the transmission line works from DC to 32 GHz. With the determined resonator radius, it has a dual mode resonance at 17.3 GHz. Since there is a strong coupling between the dual modes of the resonator with the line, the resonance frequencies are separated from each other. The filter section is formed by six ring resonators placed on both sides of the SSPP-line. The resonators disturb the surface impedance of the SSPP transmission line. At certain frequencies, the signal on the SSPP transmission line is coupled to the resonators, leading to stop bands.

Figure 3 shows the configuration of two rings coupled to the SSPP-TL and its signal flowgraph. The signal flowgraph is composed of the two similar two-port networks of the resonators, which are connected to the two-port network of the SSPP-line. Transmission of the signal around the rings is given by $a_{3}=\alpha e^{j\theta }b_{4}\left(a_{5}=\alpha e^{j\theta }b_{6}\right)$, where α and θ represent the attenuation and the phase shift of the signal going through a CCRR. θ is equal to $\frac{4\pi^{2}R_{R}}{\lambda_{g}}$. The $S_{21}$ of the line coupled with two CCRRs can be determined by:(5)$$S_{21}=\frac{b_{2}}{a_{1}}=\frac{T-\alpha \left(2KK^{*}+TT^{\prime }\right)e^{j\theta }}{\left(1-\alpha T^{\prime }e^{j\theta }\right)},$$
(6)$${\left|S_{21}\right|}^{2}=\frac{t^{2}+{\left(\alpha tt^{\prime }\right)}^{2}+4\alpha^{2}k^{4}+4\alpha^{2}tt^{\prime }k^{2}cos\left(\varphi_{t}+\varphi_{t^{\prime }}\right)-2\alpha t^{2}t^{\prime }cos\left(\theta +\varphi_{t^{\prime }}\right)-4\alpha tk^{2}cos\left(\theta -\varphi_{t}\right)}{1+{\left(\alpha t^{\prime }\right)}^{2}-2\alpha t^{\prime }cos(\theta +\varphi_{t^{\prime }})}.$$

In these two equations, $T=te^{j\varphi_{t}}$ is the self-coupling ratio of the SSPP line at CCRRs; $T^{\prime }=t^{\prime }e^{j\varphi_{t^{\prime }}}$ is the self-coupling ratio of the CCRRs; $K=ke^{j\varphi_{k}}$ is the cross-coupling ratio between the SSPP-line and CCRRs. Under the symmetrical & lossless coupling, and negligible reflections, we can assume that $t^{2}+2k^{2}=1$. The transmitted signal of one CCRR row shows attenuation on wanted frequencies; therefore, using multiple rows of CCRRs increases the resonance depth in these frequencies. The resonance condition occur when $(\theta +\varphi_{t}+\Delta \varphi )=2m\pi$, where *m* is the integer, and Δφ is a small value related to the difference of the coupled and transmitted signal phase.

Figure 4 shows the E-field distribution of the SSPP-line coupled with CCRRs at the two resonance frequencies, i.e., 9.5 GHz and 20.9 GHz. At these two resonant frequencies, the E-field of the CCRRs mostly concentrates on the circular gaps. The E-fields are mainly cancelled around the first two CCRRs, which is a good depiction of termination propagated waves at two forbidden bands.

## 3. Parametric Study

In order to illustrate the filter controllability and investigate the influence of geometrical parameters on filtering properties, four parameters have been chosen, i.e., *n* (the number of CCRRs), $W_{R}$, (the width of CCRRs), $y_{R}$ (the distance of center strip to the CCRRs center in y-direction), and $R_{R}$ (the CCRRs radius).

Figure 5a,b depicts the reflection and transmission coefficients of the structure with different numbers of CCRRs. Without the CCRR, the SSPP line works in an ultra-wideband frequency range from 1.85 to 32 GHz with an fractional bandwidth (FBW) of 178%. By introducing CCRRs, two forbidden bands are obtained. With more CCRRs, the resonances get deeper, which is in accordance to Equation (Equation 5). However, the variation of the S-parameters is minor by adding more CCRRs when *n* > 6. Therefore, to obtain a compact structure, *n* is selected to be 6. Meanwhile, $W_{R}$ plays a crucial role in the filter tunability. As shown in Figure 5c,d, by changing $W_{R}$, the second resonance changes, while the first one remains the same. By increasing the value of $W_{R}$ from 0.1 mm to 0.3 mm, the second resonance shifts from 19.6 GHz to 22 GHz, which is 11.5% with respect to the center frequency of 20.8 GHz. This parameter does not influence the depth of the resonance. In addition, the second resonance can be adjusted by changing $y_{R}$.

Figure 6a,b plots the variations of the reflection and transmission coefficients of the filter with $y_{R}$ swept from 3.55 mm to 3.75 mm. As can be seen, the resonance shifts from 21.9 GHz to 21 GHz. At last, Figure 6c,d illustrates the change of the resonance when $R_{R}$ varying from 2.3 mm to 2.5 mm. As observed from the figure, $R_{R}$ affects both the first and the second resonances. By adjusting $W_{R}$ at the same time to compensate the change of the $R_{R}$ effect on the second resonance, the first resonance can be adjusted without changing the second one. As illustrated in Figure 7, by simultaneously changing $W_{R}$ and $R_{R}$, the first resonance shifts from 9.4 GHz to 10.1 GHz (7.17% with respect to the center frequency of 9.75 GHz), while the second resonance remains unchanged.

## 4. Fabrication and Measurement

The proposed single-layer D-BRF was also fabricated and tested. A prototype was fabricated by printed circuit borad (PCB) technology. The substrate of the PCB (Rogers 4003C) is 0.208-mm with the permittivity of 3.55, and the thickness of the single-side printed copper film is 0.018 mm. The fabricated prototype and its measured results are shown in Figure 8. The total size of the fabricated D-BRF is 50.15 mm * 15 mm, which is 3.8$\lambda_{g}$ * 1.1$\lambda_{g}$ ($\lambda_{g}$ is the guided wavelength at a center frequency of 15 GHz, which is around the average center frequency of the forbidden bands). Generally, the measured results agree well with simulation, which verified the design principle. At high frequencies, it is worth mentioning that the measured parameters are different with the simulated ones. This difference may be due to the parasitic characteristics of the SMA connectors and substrate, as well as the mediocre fabrication techniques. Two forbidden bands of the D-BRF locate in X and K bands, and they can be easily shifted through redesigning the structure. The first band covers the frequency range of 8.6–11.2 GHz with a fractional bandwidth (FBW) of 26.2%. The second one works in the frequency range of 20–21.8 GHz with an FBW of 8.6%. As it was expected from the dispersion diagram of the cell, cut-off frequency of the D-BRF is 32 GHz. The first and second rejection bandwidth (below −20 dB) are 2.6 GHz and 1.8 GHz.

To obtain an intuitive understanding of the D-BRF performance, the near-electric-field distribution at different frequencies (two frequencies in the forbidden bands and three other frequencies in the pass bands) are illustrated in Figure 9. In the forbidden bands, the propagated EM waves are aborted at the location of the CCRRs. At frequencies within the pass bands, near-electric field results of the propagated waves show the EM energy transmission from the input port to the output port as if there are no CCRRs in the line.

The proposed dual band-rejection-filter is also compared with the state-of-the-art dual-band-rejection filters. According to the Table 1, compared with previous ones, the proposed filter provides deep stop bands with higher fractional bandwidth. The forbidden bands of the filter cover the frequency bands of X and K. Moreover, the return loss (RL) of the designed filter in the two stop bands are higher than −2.5 dB.

## 5. Conclusions

In this work, a new single-layer SSPP-based dual band-rejection-filter has been proposed. The design idea of this filter originates from interaction between the SSPP structure and a coupling part comprising of six CCRRs. The SSPP-cell determines the filter’s high cut-off frequency, and the filter part creates two forbidden bands. By changing some geometrical parameters of the structure, the filter has offered tunable forbidden bands with wide rejection bandwidth. The proposed filter is low-cost, easy to fabricate, and easy to integrated with other systems, which makes it a good candidate in communication systems. At last, a prototype has been fabricated and tested. The measured results agree well with the simulated results, which validated the design principle. Compared to state-of-the-arts designs, the proposed filter offers two much wider forbidden bands, which, for example, could be used to reduce the unwanted interference from satellites.

## Figures and Tables

**Figure 1 sensors-20-07311-f001:**
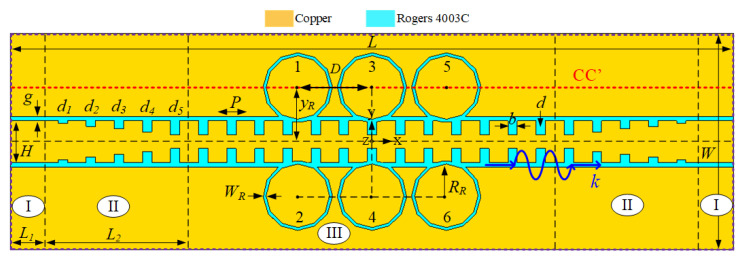
Schematic of the proposed D-BRF through interaction between the coupled circular ring resonators (CCRRs) and the Spoof Surface Plasmon Polariton (SSPP)-based transmission lines (TL).

**Figure 2 sensors-20-07311-f002:**
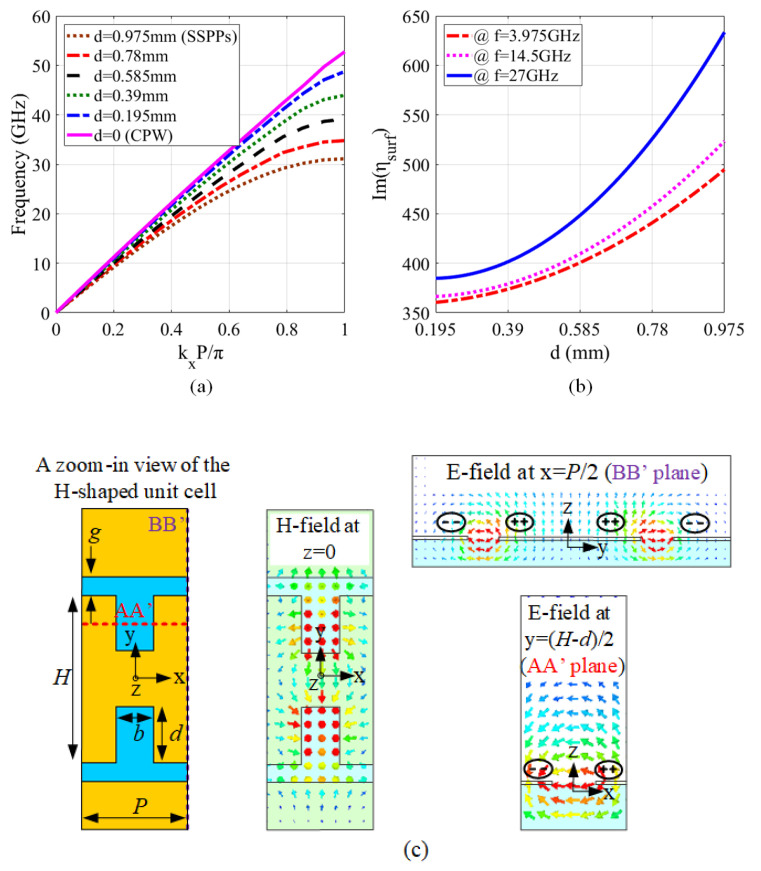
(**a**) Effect of the groove depth on dispersion curve of the cell. (**b**) The surface impedance of the cell by groove depth change in different frequencies. (**c**) The distribution of E-field & H-field at three orthogonal planes (E-field at A-A’, and B-B’, H-field at z = 0).

**Figure 3 sensors-20-07311-f003:**
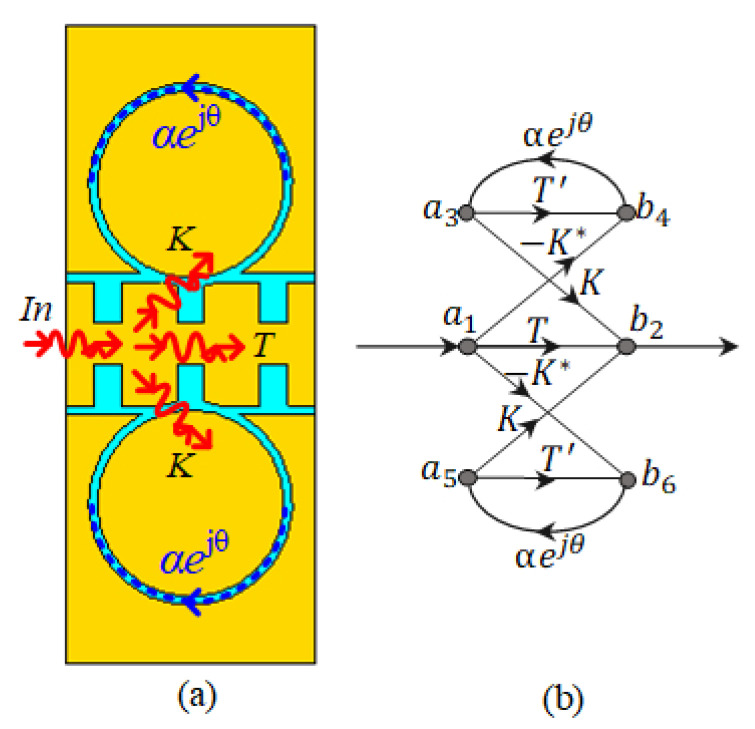
(**a**) Filter part of the D-BRF with two CCRRs coupled to the SSPP-TL. (**b**) Signal flow-graph of the filter section.

**Figure 4 sensors-20-07311-f004:**
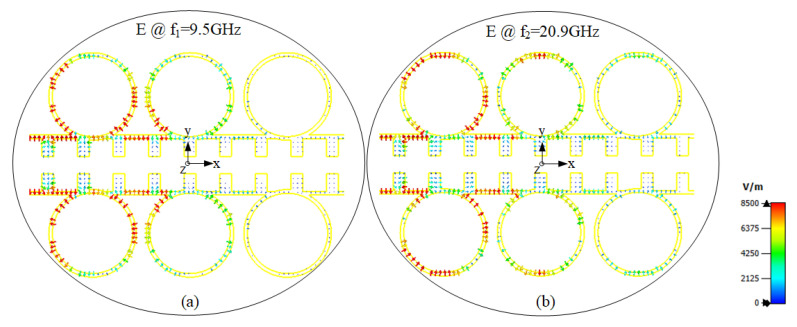
Electric field lines of CCRRs, (**a**) at the first resonance ($f_{1}$ = 9.5 GHz) and (**b**) at the second resonance ($f_{2}$ = 20.9 GHz).

**Figure 5 sensors-20-07311-f005:**
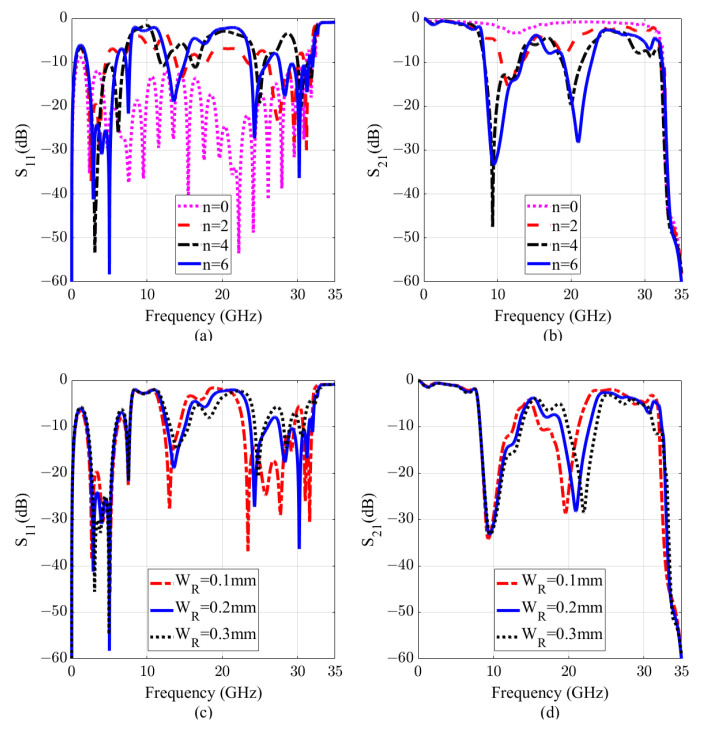
Scattering parameters of the D-BRF. (**a**,**b**) with different numbers of CCRRs (*n*). (**c**,**d**) with change of the CCRRs width ($W_{R}$). By changing the $W_{R}$, the second resonance mode can be independently controlled within the frequency range of 19.6 to 22 GHz.

**Figure 6 sensors-20-07311-f006:**
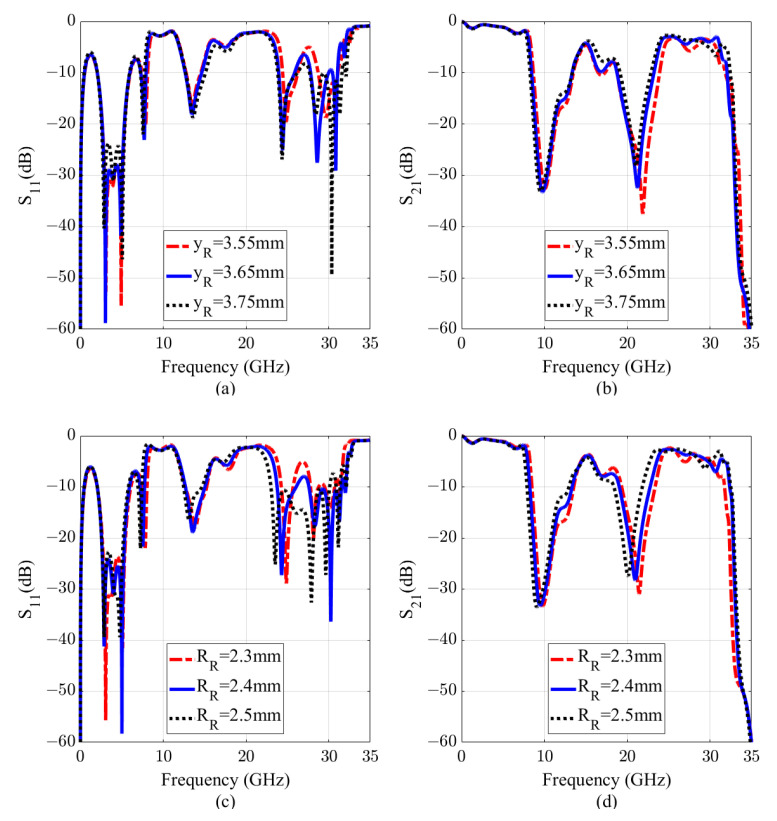
Scattering parameters of the D-BRF. (**a**,**b**) with different value of the CCRRs distance to the SSPP-TL ($y_{R}$). (**c**,**d**) with change of the CCRRs radius ($R_{R}$).

**Figure 7 sensors-20-07311-f007:**
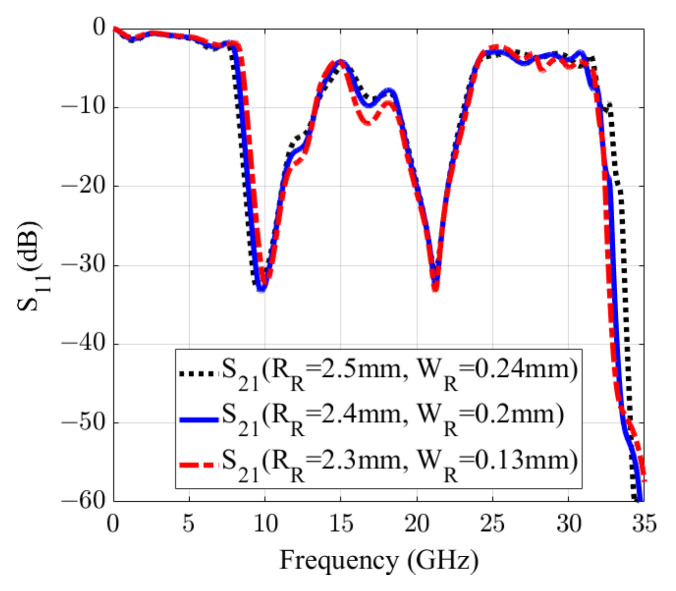
Scattering parameters of the filter and the independent movement of the first resonance by different value of the width and the radius of the resonators. The first resonance mode can be independently controlled within the frequency band from 9.4 to 10.1 GHz.

**Figure 8 sensors-20-07311-f008:**
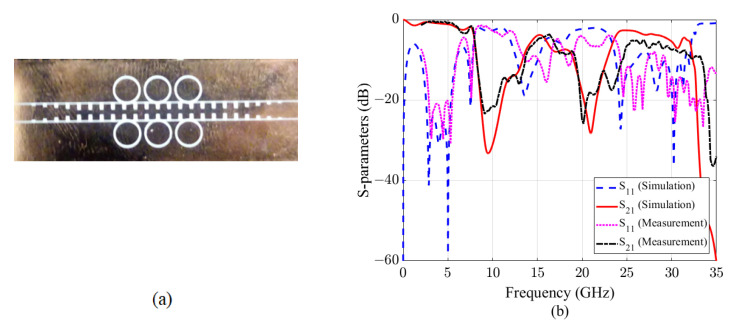
(**a**) Picture of the fabricated sample. (**b**) The simulation and experiment results of the D-BRF.

**Figure 9 sensors-20-07311-f009:**
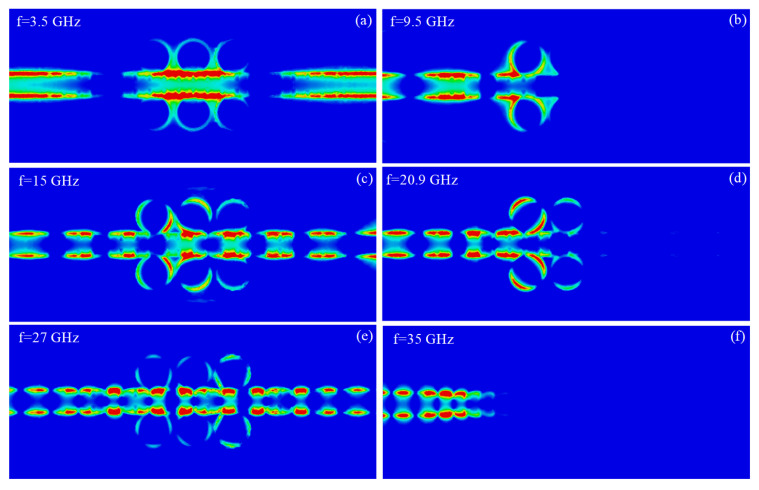
Electric-field distribution of the D-BRF, 0.2 mm above the structure, at different frequencies (in the rejection and pass bands). (**a**) 3.5 GHz, (**b**) 9.5 GHz, (**c**) 15 GHz, (**d**) 20.9 GHz, (**e**) 27 GHz, and (**f**) 35 GHz.

**Table 1 sensors-20-07311-t001:** Performances comparison of the proposed dual band-rejection-filter with traditional SSPP-based ones. The red is shown the advantages of this work, which is wide rejection bandwidth.

Ref.	SL or DL	$f_{0}$ (GHz)	FBW (%)	Depth (dB)	RL in SBs (dB)	IL in PBs (dB)	Tunability
[30]	SL	8.21/10.4	NBs	−34/−31	NA	2.8/2.6/2.6	Yes
[31]	SL	5.36/9.32	1.61/1.29	−30/−25	NA	3/2.3/2.4	No
[32]	DL	4.04/4.5	NBs	−7.5/−8	−5/−5	0.3/0.3/0.3	Yes
[33]	SL	7.65/9.47	NBs	−15/−17	−8/−8	2.5/2.3/2.6	Yes
[35]	DL	9.15/11.55	5.5/9.5	−40/−50	NA	0.8/1.2/1.9	Yes
[36]	DL	18.6/22.6	5.2/6	−43/−60	−2.7/−2.4	0.9/2.7/1.3	Yes
[37]	SL	6.4/6.8	NBs	−20/−18	NA	2/2.5/2.8	Yes
Here	SL	9.5/20.9	**26.2/8.6**	−32/−25	−2.5/−2.5	1.5/6.8/3.8	Yes

SL: single layer; DL: double layer; FBW: fractional bandwidth; RL: return loss; SBs: stop bands; IL: insertion loss; PBs: pass bands; NB: narrow bands; NA: not assigned.

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
