# Peer review of "A Spoof Surface Plasmon Polaritons (SSPPs) Based Dual-Band-Rejection Filter with Wide Rejection Bandwidth"

_sensors, 2020, doi:10.3390/s20247311_

Round 1
Reviewer 1 Report
The authors have designed and fabricated a single-layer dual band-rejection-filter based on Spoof Surface Plasmon Polaritons (SSPPs). They claimed that the proposed filter possesses tunable rejection bandwidth and wide stop bands. In my opinion, this manuscript is well written and the results are interesting for the readers and deserve publication in this Journal after major revision.
Q1:, I suggest the term “rejection bands” can be replaced by “forbidden bands” throughout this manuscript.
Q2: Authors use three pairs of CCRRs on the two sides to design the filter. Please describe the difference if CCRRs are only used on one side.
Q3: A color scale bar with intensity should be added in Fig. 4.
Q4: In section 4, please briefly elucidate the fabrication method of the proposed structure.
Q5: I suggest to move the inset of Fig. 8 to the top of Fig. 8.
Q6: The simulation method and setting used in this work should be mentioned in the text. Authors should cite the website if the commercial software (including the version) is used.
Q7: The references used in this work should be improved. In order to be beneficial for the readers to know the mechanism and other approaches regarding the properties associated with dual-band forbidden filter (e.g., Results in Physics, 103116 (2020) and Nanomaterials 10 (3), 493 (2020)), spoof surface plasmon polaritons (e.g., J. Phys. D: Appl. Phys. 50, 125302 (2017)) and ultrawide bandgap (e.g., Nanomaterials 10, 2030 (2020)), authors should cite the above-mentioned literature.
Reviewer 2 Report
The manuscript treats the design, analysis, fabrication, and testing of a dual-band-rejection filter that is based on spoof surface plasmon polaritons and circular ring resonators.
The paper is well presented and the results show filters with wide rejection bands.
As a recommendation the term ¨tunable¨ in the title and the characterization of the ring resonator as complementary seem to me being inappropriate. Where is complementarity by referring to the ring resonator?
Also, the abbreviation CPW is used before being defined, although the abbreviation is a well known term.
In conclusion, I recommend the manuscript for publication after the above changes are considered.
Round 2
Reviewer 1 Report
The authors have revised the manuscript according to my comments and the manuscript can now be accepted for publication.
Author Response
Thank you very much for your positive comments and constructive suggestions in the previous round of revision.